# Impact of culture on refugee women's conceptualization and experience of postpartum depression in high-income countries of resettlement: A scoping review

**Saarah Haque[1], Mary Malebranche[2]** *

**1** University of Toronto, Toronto, Ontario, Canada, **2** Department of Medicine, Cumming School Medicine, University of Calgary, Calgary, Alberta, Canada

☯ These authors contributed equally to this work.
* mary.malebranche@ucalgary.ca

## Abstract

**Data Availability Statement:** All relevant data are within the paper and its Supporting Information files.

### Background

The global refugee population has reached a staggering 25.9 million. Approximately 16% of global refugees resettle in high-income countries which are often culturally very different from their home countries. This can create cross-cultural challenges when accessing health services, leading to inappropriate assessments, diagnoses and treatments if cultural background is not factored in. The impact of culture on the conceptualization and experience of postpartum depression (PPD) amongst migrant women has received growing attention in recent years, however, a specific focus on refugee and asylum-seeking women is lacking. Given the unique mental health challenges refugee women face, it is hypothesized that the interplay between culture and postpartum depression amongst refugee women may differ from other migrant women. Therefore, a scoping review was conducted to characterize what is known about the impact of culture on the conceptualization and experience of PPD in refugee women resettled in high-income countries.

### Methods and findings

This study was conducted as a scoping review in accordance with the Joanna Briggs Institute's Methodology for Scoping Reviews. A systematic search of studies addressing the relationship between culture and postpartum depression amongst refugee women (including asylum-seeking women) resettled in high-income countries was conducted across 6 databases including MEDLINE, PsycINFO and SOCINDEX between June 2018 and August 2019. A total of 637 articles were found. Studies were eligible if they focused on refugee women who had a pregnancy during forced migration or upon resettlement in a high-income country and focused on the impact of culture on women's conceptualization and/or experience of PPD. Eight studies met inclusion criteria and were included in the final analysis, the majority of which were qualitatively driven. Four key themes emerged: 1) there are diverse conceptualizations and experiences of postpartum depression amongst refugee women; 2)

**Funding:** The authors received no specific funding for this work.

**Competing interests:** The authors have declared that no competing interests exist.

mental health stigma has a significant impact on women's conceptualizations and experiences of postpartum depression and help-seeking behaviors; 3) cultural traditions and social support play protective roles in postpartum mental wellbeing; and, 4) host culture has a significant influence on the pregnancy and postpartum experience of refugee women. The overall themes align with those seen in the literature on migrant women in general, however significant research gaps remain.

## Conclusion

The studies identified through this scoping review provide a rich description of the significant impact culture has on the conceptualization and experience of postpartum depression among refugee women resettled in high-income countries. Though overall themes align with those seen in the literature on migrant women in general, further research is needed to better characterize how culture impacts refugee women's experiences of PPD as a distinct sub-group of migrant women.

## Introduction

At the end of 2018, the global refugee population reached a staggering 25.9 million [1]. Each year, in increasing numbers, people from around the world are forced to flee their home countries to seek refuge from persecution, conflict, violence, and human rights violations [1]. The UNHCR define refugees as people "who have fled war, violence, conflict or persecution and have crossed an international border to find safety in another country", while asylum seekers (also known as refugee claimants) are people who have made in-country claims for asylum that have yet to be processed [2,3]. The majority of refugees resettle in neighboring low- and middle-income countries but an estimated 16% eventually resettle in high-income countries including the United Kingdom (UK), Australia and Canada–countries that are often geographically and culturally very different from their countries of origin [4–8].

Almost half of global refugees are women, many of childbearing age, who experience pregnancy and give birth during their migration journeys or upon resettlement [4]. Giving birth during a time of forced migration and early resettlement increases refugee women's vulnerability to numerous adverse maternal, obstetric, and newborn outcomes [9]. One such adverse outcome is postpartum depression (PPD). Studies demonstrate a 2–5 times greater risk of refugee women resettled in high-income countries developing PPD compared to non-migrant women [10,11].

Postpartum depression is defined in the Diagnostic and Statistical Manual of Mental Disorders, 5th Edition (DSM-5) as a depressive episode that occurs within 12 months of childbirth. PPD has its origins in a Western biomedical model of disease that emphasizes individual biological and psychological factors [12]. If untreated, PPD can have negative effects on the mother, child, and the whole family unit [13]. Though it is recognized that PPD is also influenced by social and cultural factors, these factors are often under-recognized in the assessment and management of PPD in refugee women [14]. In this particular population, the conceptualization and experience of PPD may differ significantly from Western views that predominate in their high-income country of resettlement [14].

The impact of culture on the conceptualization and experience of PPD in migrant women in general has received growing attention in the literature [14–16]. However, studies tend to group all subtypes of migrants together and consequently lack special attention to the sub-

population of refugee women. Refugee women, however, may face unique challenges compared to women with other migration journeys due to the forced nature of their migration, as well as other factors that have yet to be fully understood [17–19]. An examination of available evidence regarding the impact of culture on the conceptualization and experience of PPD which focuses on refugee women resettled in high-income countries is needed. A focus on refugee women resettled in high-income countries has been chosen for two main reasons. Firstly, refugee women resettled in high-income countries face a high risk for developing PPD which drives our interest in better understanding this phenomenon [10,11]. Secondly, increasing numbers of refugee women of childbearing age are resettling in high-income countries [20]. As such, it is imperative we gain a better understanding of how culture impacts the ways in which refugee women understand and describe their postpartum experiences when resettling in high-income countries, as well as how our own culture (and interface between our culture and that of our patients) impacts how we assess, diagnose and treat PPD in refugee women.

As such, the objective of our study is to provide a broad and inclusive overview of available evidence regarding the impact of culture on the conceptualization and experience of PPD amongst refugee women resettled in high-income countries. Such an overview will enable us to clarify key concepts and findings in the literature, while also identifying important knowledge gaps to guide future research to inform clinical care in high-income countries by improving the clinical detection and management of PPD amongst refugee women. In order to achieve these research objectives a scoping review was conducted in accordance with JBI's Methodology for Scoping Reviews.

## Methods

This review was conducted in accordance with the Joanna Briggs Institute's Methodology for Scoping Reviews [21] whereby the objectives, inclusion criteria, and methods were specified in advance and documented in an a priori scoping review protocol. As an inherently flexible, inclusive and iterative methodology, it was felt that a scoping review would ensure all relevant literature was captured.

### Eligibility criteria

**Participants.** Our study population of interest was women identified as refugees, refugee claimants or asylum seekers who had a pregnancy during forced migration or upon resettlement. The UNHCR define refugees as people "who have fled war, violence, conflict or persecution and have crossed an international border to find safety in another country", while asylum seekers (also known as refugee claimants) are people who have made in-country claims for asylum that have yet to be processed [2,3]. Refugee claimants and asylum seekers are classified as undocumented migrants [22]. Undocumented migrants also include those who overstay their visas, and those who are not seeking refugee status but stay within a country without permission–these individuals were not included in the scoping review [22]. For the purposes of this study, the term "refugee" will refer to refugee, refugee claimant and asylum-seeking women. There were no restrictions on women's age, parity, country of origin or time from resettlement to study involvement. The definition of PPD diagnosis was inclusive to women who had a previous diagnosis of PPD confirmed by screening tools or clinical depression. We also included studies where women with previous pregnancies discussed their understanding of PPD if the study included women with PPD symptomatology. Studies with immigrant or migrant women as the sole focus or studies in which the analysis did not include a sub-analysis on refugee women were excluded.

**Concept.** The phenomena of interest were the impacts refugee women's native culture, as well as the host country's culture (i.e. the cross-cultural impacts), have on refugee women's conceptualization and experience of PPD in their country of resettlement. Here, we define cross-culturalism drawing from the definition used in the discipline of cross-cultural psychology whereby cross-culturalism is the "similarities and differences in behavior among individuals who have developed in different cultures." [23]. Cultural factors may be seen as harmful, protective or neutral in relation to their impact on refugee women's mental health and wellbeing in the postpartum period. Therefore, studies that addressed cultural factors in the postpartum period were included, using the World Health Organization definition of culture: "the set of distinctive spiritual, material, intellectual and emotional features of society or a social group . . . [which] encompasses, in addition to art and literature, lifestyles, ways of living together, value systems and traditions and beliefs." [24]. Studies that focused only on the prevalence of PPD in certain ethnic groups were excluded. The outcome measure for our research question was the refugee woman's conceptualization and/or experience of PPD.

**Context.** The specific context of interest was refugee women resettled in high-income countries as defined by the World Bank Atlas method [25]. No restrictions were placed on health care setting of the study (i.e. acute care, primary care of the community).

**Study sources.** Primary research, review articles and grey literature including policy documents, government reports, conference proceedings, abstracts, dissertations/theses, and clinical practice guidelines were included in the scoping review. Clinical practice guidelines from the last 5 years were considered. Excluded sources included books, opinion articles, and informal communications. Refer to Table 1 for a summary of eligibility criteria.

## Search strategy

The search strategy was developed as a collaborative effort between authors (SH, MM), with guidance from a health sciences research librarian. The literature search was built around the 3 core concepts of "refugee", "postpartum depression" and "culture" using search terms outlined in Table 2. Comprehensive searches were completed in 6 databases: MEDLINE (Ovid), PsycINFO (Ovid), SOCINDEX, CINAHL, Sociological Abstracts, and Social Sciences Citation Index. Search strategies using Medical Subject Headings (MESH) and text words were created for each database (S1 Table). Specific search terms were adjusted as new terms were identified throughout the literature review process. The 3 core concepts were combined using Boolean operators "OR" and "AND".

**Table 1. Eligibility criteria overview.**

|  | Inclusion | Exclusion |
|---|---|---|
| **Report Characteristics** | | |
| Language | No restrictions. | |
| Literature Sources | Primary research, grey literature (policy documents, government reports, conference proceedings, abstracts, dissertations/theses, clinical practice guidelines, ongoing studies), and review articles. | Books, opinion articles, and informal communications. |
| **Study Characteristics** | | |
| Sample population | Studies that focused on refugee, refugee claimant, and asylum-seeking women who had a pregnancy or gave birth after resettlement. | Studies that focused on immigrant or migrant women in general. Studies that included women with pre-existing mental illnesses. |
| Location | High-income countries of resettlement. | |
| Outcome | Conceptualization and/or experience of PPD (e.g. positive, negative, neutral) | |
| Study design | No restrictions | |
| Focus | Impact of culture, both native culture and the culture of the host country. | Prevalence of PPD in specific ethnic groups. |

**Table 2. Search terms.**

| | |
|---|---|
| **Refugee status** | refugee* or refugee claimant* or resettled refugee* or convention refugee* or migrant* or asylum seek* or asylum-seek* or asylum claimant* or asylum-claimant* or protected person* or displaced person* or stateless person* or person* without status or undocumented person* |
| **Postpartum depression** | postpartum depression or post-partum depression or postpartum mental health or post-partum mental health or postnatal depression or post-natal depression or postnatal mental health or post-natal mental health or antenatal depression or ante-natal depression or antenatal mental health or ante-natal mental health or perinatal depression or peri-natal depression or perinatal mental health or peri-natal mental health or maternal depression or maternal mental health or puerperal depression or puerperal mental health |
| **Culture** | cultur* or sociocultur* or socio-cultur* or religion* or religious or belief* or value* or social or support* or societ* or norm* or tradition* or ritual* or language* or gender or acculturation or assimilation or assimilate* or resettle* |

Following the database search, a citation search, reference list search, and grey literature search was carried out. Citation searching was conducted using Scopus and Google Scholar. A primary grey literature search was carried out using Google Scholar. Further, the grey literature search included looking at relevant clinical practice guidelines, publications by civil society groups (including obstetrics and gynecology, midwifery, and nursing groups in high-income countries) and assessing ongoing research using PROSPERO.

The search was initially conducted in June 2018 but was updated based on our evolving knowledge in this area and finalized in August 2019. No restriction was placed on study methodology, language of publication or date of publication.

**Study selection.** All literature identified through the search was uploaded to Covidence, an online systematic review management program. In Covidence, all articles were subject to a title and abstract review based on the predefined eligibility criteria. This was followed by a full text review of eligible studies. Disagreements were resolved through discussion between both reviewers (SH, MM). Each step was conducted in parallel by two independent reviewers (SH, MM). Studies that met eligibility criteria following the full-text review were included in the final scoping review and subject to data extraction.

**Data extraction.** Relevant data were extracted from each included study by two independent reviewers (SH, MM) using a predefined data extraction form (S2 Table). This form was revised throughout the literature review process as authors became more familiar with search results, adding further data extraction fields when appropriate.

**Quality assessment.** As per agreed upon scoping review methodology, no rating of quality or level of evidence for included studies was conducted [21].

## Results

A total of 637 articles were identified through the search strategy. After removing duplicates, 337 articles were included in the title and abstract screen and 45 in the full text screen. A total of 8 studies met eligibility criteria and were included in the final review (Fig 1). No studies identified through the search of grey literature met inclusion criteria.

The majority of included studies were qualitatively driven. Ahmed et al. (2017) and Russo et al. (2015) had a qualitative mixed methods design including focus groups, interviews, and questionnaires [26,27]. Stapleton et al. (2013) also took a mixed methods approach involving both quantitative and qualitative data gathered from hospital databases, chart audits, surveys, and interviews [28]. Ussher et al. (2012) conducted a qualitative study that included five semi-structured focus groups [29]. The study by Davey (2013) was a master's thesis that included community based participatory research [30]. Two studies were not primary research: Brown-

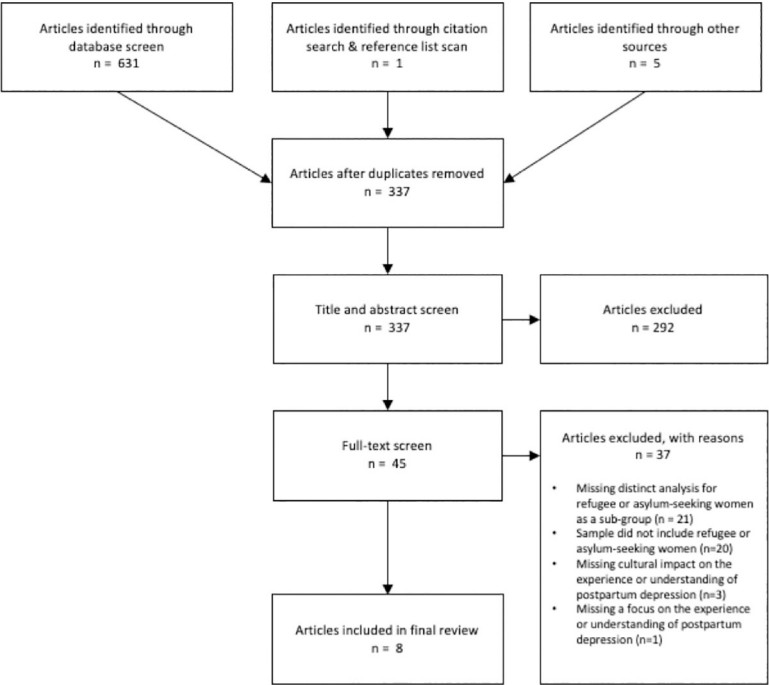

**Fig 1. PRISMA flowchart demonstrating selection of articles.**

Bowers et al. (2015) wrote a theoretical article using a critical health psychology framework [12] and Asif et al. (2015) wrote a general review paper [31]. Lastly, Nakash et al. (2016) conducted a quantitative, cross-sectional study [32]. All studies focused primarily on the perspectives and experiences of refugee women themselves. Stapleton et al.'s study (2013) also explored the perspectives and experiences of health care providers (i.e. maternity clinic staff) and community stakeholders [28]. The included studies were conducted across notably diverse disciplines including Psychology, Nursing, Midwifery, Medicine, Public Health, and Anthropology. All studies were published within the past 8 years (between 2012 and 2017).

The studies were conducted in the following high-income countries of resettlement: Australia (n = 3), Canada (n = 3), the United Kingdom (n = 1), and Israel (n = 1). The refugee and asylum-seeking women who participated in the studies represented diverse ethnic and cultural backgrounds including Eritrean (n = 38), Afghani (n = 38), Karen (n = 26), Assyrian (a Christian indigenous population living in areas of modern-day Iraq, Iran, Turkey, and Syria) (n = 16), Syrian (n = 12), and Bhutanese (n = 12). Further, a number of refugee women from countries including Somali, Sudan, Burundi, Afghanistan, and Liberia participated in Stapleton et al.'s mixed methods study (n = 250) [28]. Both married and unmarried women of diverse educational and socioeconomic backgrounds with varying time since resettlement (ranging from under 6 months to 6 years) were represented across studies in which demographic data was provided [26,27,30,32].

The majority of studies focused on assessing the cultural conceptualizations (sometimes described as "perceptions" or "constructions") and experiences of pregnancy and the postpartum period for resettled refugee women from a mental health perspective [26,27,29–32]. Brown-Bowers et al. uniquely explored the current problematization of PPD in refugee women, offering an alternative approach for understanding and addressing this problem while Nakash et al. focused on the impact of acculturation on the mother-infant bond amongst

refugee women [12, 32]. Stapleton et al. explored the short-comings of applying the Edinburgh Postnatal Depression Scale (EPDS) to the care of recently resettled, culturally diverse, refugee and asylum-seeking women given its basis within the Western biomedical model of disease [28]. Refer to Table 3 for a summary of the 8 included studies.

## Key themes

Results of this scoping review are presented according to 4 key themes that emerged throughout the review. The themes, summarized in Table 4, were: 1) diversity in cultural conceptualizations and experiences of postpartum depression; 2) impact of mental health stigma on women's conceptualizations and experiences of postpartum depression and help-seeking behaviors; 3) protective role of cultural traditions and social support; and, 4) influence of host culture on the experience of pregnancy and postpartum depression.

**1. Diversity in cultural conceptualizations and experiences of postpartum depression.** Among the various studies that focused on the cultural conceptualizations of postpartum depression in resettled refugee and asylum-seeking women, a diversity of views were found, ranging from women having no cultural concept of postpartum depression to expressing an understanding of the condition and disclosing their own experiences with PPD [29, 30].

In Ahmed et al. (2017), Syrian refugee women in Canada endorsed having heard of postpartum depression but reserved the term "depression" for extreme cases as it means "being sick and requiring treatment" [26]. They themselves did not endorse any depressive symptoms, sharing a belief that Syrians experience less PPD than other women due to their cultural view of birth as a celebratory time [26].

Alternatively, in the study by Davey (2013), Bhutanese refugee women lacked a cultural conceptualization of postpartum mental distress altogether [30]. This is attributed to the socially determined concept of wellness in Bhutanese culture whereby one's sense of health is a reflection of the quality of one's social relationships [30]. Therefore, it is taboo to outwardly express personal distress as it is unacceptable to burden other members of the community with personal issues [30]. Further, study participants explained that suffering is perceived as a normal part of life in Bhutan and therefore, women are expected to endure hardships associated with birth and motherhood without complaint [30].

Ussher et al. (2012), in studying both Karen and Assyrian refugee women resettled in Australia, found contrasting cultural conceptualizations of PPD amongst these two culturally distinct groups [29]. While Karen refugee women described PPD as an "alien experience" for women who had a healthy baby, Assyrian women had a clear understanding of PPD with some participants even disclosing their own experiences with severe depression after birth [29]. Two Assyrian study participants further described having spoken to their husbands and family about their depressive symptoms in the postpartum period [29].

Stapleton et al. (2013) also note that cross-cultural differences in understandings of emotional states will influence not only how postpartum emotions are described but also how symptoms are reported [28]. This is further addressed in the review by Asif et al. (2015) in which it is noted that somatic symptoms such as headaches, non-specific chest pain and abdominal pain are common presentations for underlying emotional distress amongst refugee and asylum-seeking women in the UK [31].

In Stapleton et al. (2013), study participants postulated that PPD was essentially a Western illness, suggesting that refugee women from non-Western cultures may lack a framework for understanding the rationale behind administering a tool like the Edinburgh Postnatal Depression Scale, for example [28]. Authors then go on to provide a compelling overview on this theme stating that despite the internationalization of the term "depression" it is unlikely that

**Table 3. Summary of included studies in alphabetical order by first author.**

| Authors, Year & Title | Country | Methodology | Sample Size | Participant Characteristics | Study Objective(s) | Emerging Themes | Key Findings Related to the Impact of Culture on PPD | Recommendations for Future Research |
|---|---|---|---|---|---|---|---|---|
| Ahmed et al. 2017 Maternal depression in Syrian refugee women recently moved to Canada: a preliminary study. | Canada | Mixed methods *Included*: questionnaires, EPDS* and focus group discussion | N = 12 | Syrian refugee women resettled in Saskatoon, either pregnant or 1 year postpartum | To explore Syrian refugee women's' experiences of expecting or having a baby after resettlement from a mental health perspective. | • Cultural conceptualizations and experiences of PPD • Impact of stigma on conceptualizations, experiences, and help-seeking • Protective role of traditions and social support • Influence of host culture and the Western biomedical model of disease | • Participants had many misconceptions about maternal depression. • Social support and spiritual practices during pregnancy, birth, and postpartum seen as protective for mental wellbeing. • Stigma and privacy concerns identified as significant barriers to seeking or accessing mental health services. | • Increased awareness and education is needed about symptoms, effects, causes and treatment of maternal depression amongst Syrian refugees and their families. • Reuniting women with their families and engaging them in culturally appropriate support programs may improve mental health outcomes. |
| Asif et al. 2015 The obstetric care of asylum seekers and refugee women in the UK. | United Kingdom | Review | N/A | Asylum seekers & refugee women | Gain knowledge of how to provide good antenatal care for this group of vulnerable women. To be able to describe medical, sexual & psychosocial issues that face pregnant asylum seekers & refugee women. | • Diversity in cultural conceptualizations and experiences of PPD • The impact of mental health stigma on women's conceptualizations and experiences of PPD and help-seeking behaviors • The influence of host culture and the Western biomedical model of disease on the experience of pregnancy and PPD | • PPD is more common in women who are physically and culturally displaced from their support systems during pregnancy/ birth. There may be cultural differences in the way asylum-seekers manifest psychological symptoms (i.e. somatic complaints) and seek help. Personal shame, stigma, and reluctance to disclose emotional problems may delay diagnosis. FGM* may contribute to poor ante- and postnatal mental health. | • Ensure HCPs^ are aware of cultural barriers that exist when pregnant asylum seekers access antenatal care. • HCPs should recognize that asylum-seeking women are a high-risk group for PPD and provide close observation and individualized care. |
| Brown-Bowers et al. 2015 Postpartum depression in refugee and asylum-seeking women in Canada: A critical health psychology perspective. | Canada | Theoretical article using a critical health psychology framework | N/A | Asylum seekers & refugee women | "To describe the Canadian context of refugee and asylum-seeking women, problematizing current dominant psychological conceptualizations of PPD and offering an alternative paradigm for understanding and addressing this problem." | • The impact of mental health stigma on women's conceptualizations and experiences of PPD and help-seeking behaviors • The influence of host culture and the Western biomedical model of disease on the experience of pregnancy and PPD | Current individualized ways of conceptualizing and addressing postpartum distress in refugee women that neglect their economic, political and sociocultural context, are problematic. Cultural differences in conceptions of motherhood between native country and host country, negative views of refugees and asylum seekers, as well as the Western biomedical approach to birth, likely contribute to postpartum distress. | • A socioecological framework may be a useful tool for generating solutions that target social conditions that extend beyond the individual that are relevant to refugee and asylum-seeking women. • Midwives are uniquely positioned to be part of a system of solutions that mitigate the postpartum distress of refugee and asylum-seeking women. |

*(Continued)*

**Table 3.** (Continued)

| Authors, Year & Title | Country | Methodology | Sample Size | Participant Characteristics | Study Objective(s) | Emerging Themes | Key Findings Related to the Impact of Culture on PPD | Recommendations for Future Research |
|---|---|---|---|---|---|---|---|---|
| Davey. 2013 The intersection between culture and postpartum mental health: An ethnography of Bhutanese refugee women in Edmonton, AB. | Canada | Dissertation Community based participatory research *Included*: semi-structured interviews and focus groups. | N = 12 | Bhutanese refugee women with a reproductive history | To assess Bhutanese refugee women's own perceptions and understandings of their postpartum wellness. To understand the intersections between selfhood, health, gender, family and community in women's culturally-mediated responses to childbirth and how this changes with migration. | • Diversity in cultural conceptualizations and experiences of PPD • The impact of mental health stigma on women's conceptualizations and experiences of PPD and help-seeking behaviors • The protective role of cultural traditions and social support during pregnancy and postpartum • The influence of host culture and the Western biomedical model of disease on the experience of pregnancy and PPD | The general lack of postpartum distress experienced by Bhutanese women can be explained by their socially-determined conceptualization of wellness and the cultural taboos of expressing personal distress. Gendered experiences that suffering is a part of normal life may contribute to resilience in the postpartum period. Traditions, rituals, and social support have a protective effect on mental wellbeing postpartum. Increased individualism and focus on personal introspection in Canadian culture may contribute to increased recognition and expression of negative postpartum feelings. | • Future research should focus on understanding women's own cultural backgrounds and individual experiences as well as continue to establish the role of cultural variables in postpartum mental health issues. • Research needs to consider cultural determinants of health including cultural models of mental and mental health. • Studies should consider men's postpartum experiences and mental health outcomes following forced migration. |
| Nakash et al. 2016. The association between postnatal depression, acculturation and mother-infant bond among Eritrean asylum seekers in Israel. | Israel | Quantitative, cross-sectional *Included*: questionnaires, Mother's Lifestyle Scale, bicultural involvement and adjustment scale, EPDS and Maternal Postnatal Attachment Scale (MPAS) | N = 38 | Eritrean asylum-seeking women within 6 months of delivery. | To assess the association between PPD, acculturation and the mother-infant bond among Eritrean asylum seekers in Israel. | • The influence of host culture and the Western biomedical model of disease on the experience of pregnancy and PPD | Higher mother-infant attachment associated with strong association with Eritrean culture and supportive relationship with partner. Lower mother-infant attachment associated with higher degree of acculturation to Israeli culture and higher EPDS score. High EPDS scores (>13) found in this cohort (81.6%). | • The current individualized approach (individual level risk factors and individual level therapies) to PPD may not be appropriate for forced migrants. A more socioecological framework is needed. • Important to establish coordinated efforts in receiving countries to screen, prevent and treat PPD in asylum-seeking women. • International medical and non-medical communities should increase awareness of this issue. |

(*Continued*)

**Table 3.** (Continued)

| Authors, Year & Title | Country | Methodology | Sample Size | Participant Characteristics | Study Objective(s) | Emerging Themes | Key Findings Related to the Impact of Culture on PPD | Recommendations for Future Research |
|---|---|---|---|---|---|---|---|---|
| Russo et al. 2015 A qualitative exploration of the emotional wellbeing and support needs of new mothers from Afghanistan living in Melbourne, Australia. | Australia | Qualitative study using feminist and sociocultural frameworks *Included*: 2 focus group discussions and 10 in-depth, semi-structured interviews | N = 38 | Afghan refugee women > 18 years who had given birth in Australia and had at least one child < 5 years | To explore the experience of pregnancy, birth and early motherhood in Afghan refugee women resettled in Melbourne To gain insights into experiences that positively and negatively impacted their emotional wellbeing throughout pregnancy and postpartum. | • The impact of mental health stigma on women's conceptualizations and experiences of PPD and help-seeking behaviors • The protective role of cultural traditions and social support during pregnancy and postpartum • The influence of host culture and the Western biomedical model of disease on the experience of pregnancy and PPD | A lack of social support, reduced connection to native culture and stigma of mental illness had negative impact on emotional wellbeing postpartum. Resistance towards discussing emotional wellbeing with HCPs was identified. Healthy relationships with one's partner, remaining connected to religion/prayer and passing down cultural ways to children had a positive impact. Positive interactions with the host culture's healthcare system and health professionals benefit refugee women postpartum. "Enhanced connection" (relational, social, cultural) is a key factor in ensuring emotional wellbeing postpartum. | • There is a need for innovative community-based models to support maternal mental health of Afghan women. • Further research into the extent and nature of Afghan women's mental health needs using both standardized measures and further qualitative inquiry would be beneficial. • Increasing social and community connection through group programs, partaking in education and employment are strategies women can use to support their mental health. • There is a need to develop a greater understanding of culturally sensitive strategies to assist Afghan refugee men during the transition to fatherhood. |
| Stapleton et al. 2013 Lost in translation: staff and interpreters' experiences of the Edinburgh Postnatal Depression Scale with women from refugee backgrounds. | Australia | Mixed methods *Included*: hospital databases, clinical chart audits, surveys, one-on-one interviews | Focus groups: N = 18 refugee women, 3 staff, 5 community stakeholders Chart audit: N = 190 Interviews: N = 3 staff Surveys: N = 42 service users, 147 maternity personnel. | Pregnant refugee women attending a dedicated refugee clinic in a maternity hospital. (Included Somali, Sudanese, Afghan, Burundi & Liberian refugee women.) Clinic staff and community stakeholders. | To explore the cross-cultural application of the EPDS[#] and the difficulties associated with administration to women from refugee backgrounds. | • Diversity in cultural conceptualizations and experiences of PPD • The impact of mental health stigma on women's conceptualizations and experiences of PPD and help-seeking behaviors • The protective role of cultural traditions and social support during pregnancy and postpartum • The influence of host culture and the Western biomedical model of disease on the experience of pregnancy and PPD | • The predominant individualistic Western biomedical perspectives and related language that inform the EPDS may not be applicable to women from collective "we" cultures. • Stigma of mental illness and varying concepts of confidentiality prevents disclosure and help-seeking for fear it might embarrass, shame or reflect negatively on her husband and family. • The absence of community bonds and close kin including cultural rituals and ceremonies can contribute to PPD in refugee women. | • Healthcare professionals require training to provide culturally competent and respectful service • The merits of routinely applying screening tools across culturally diverse populations must be critically assessed before they are adopted in practice. • Authors call for an in-depth examination into how multidimensional, cross-cultural definitions might be incorporated into future iterations of the EPDS. |

*(Continued)*

**Table 3.** (Continued)

| Authors, Year & Title | Country | Methodology | Sample Size | Participant Characteristics | Study Objective(s) | Emerging Themes | Key Findings Related to the Impact of Culture on PPD | Recommendations for Future Research |
|---|---|---|---|---|---|---|---|---|
| Ussher et al. 2012 Purity, privacy and procreation: Constructions and experiences of sexual and reproductive health in Assyrian and Karen women living in Australia. | Australia | Qualitative *Included*: 5 focus groups (2 Assyrian, 3 Karen) | N = 42 (16 Assyrian, 26 Karen) | Assyrian and Karen refugee women resettled in Western Sydney | To explore the constructions and experiences of reproductive and sexual health in Assyrian and Karen refugee women resettled in Australia | • Diversity in cultural conceptualizations and experiences of PPD • The impact of mental health stigma on women's conceptualizations and experiences of PPD and help-seeking behaviors | • There is a high value placed on motherhood in Assyrian and Karen women's culture which has negative implications for mothers who are unhappy after childbirth and those who experience PPD. • Views of PPD differed significantly between Karen and Assyrian women. • Karen: PPD an alien experience for women with a healthy baby. • Assyrian: many had heard of PPD and identified cultural stigma as a deterrent for help-seeking. • Acculturation to Western gendered roles have negatively impact postpartum emotional wellbeing. • Reproductive and sexual health needs of culturally and linguistically diverse women are not always being addressed with potential consequences for physical and psychological well-being. | • There should be more education and awareness in health care professionals when approaching PPD in culturally diverse populations • Further intersectional research is needed to examine the interaction between gender, culture, and other categories of difference in women's lives, and the outcomes of these interactions in terms of reproductive and sexual health. • Future research should explore the association between specific migration experiences and experiences of violence and trauma and women's current constructions of sexual and reproductive health. |

*FGM = Female Genital Mutilation

^HCPs = Health Care Providers

#EPDS: Edinburgh Postnatal Depression Scale

meanings across cultures will be shared [28]. This view is clearly exemplified by the diversity of cultural conceptualizations of PPD held amongst the refugee and asylum-seeking women who participated in the studies included in this review.

**2. The impact of mental health stigma on women's experience of postpartum depression and help-seeking behaviors.** The stigma that surrounds mental illness amongst refugee and asylum-seeking women from diverse cultural backgrounds was a prominent theme that emerged across 7 of the 8 included studies (all except Nakash et al, 2016). Refugee and asylum-seeking women in these studies commonly expressed how stigma of mental illness impacted their willingness to disclose emotional problems or seek help from family, community members or health care providers in the postpartum period.

In Ahmed et al. (2017), it was found that mental illness is perceived negatively in Syrian culture leading women to describe their postpartum emotional states as being "bored" or "tired" but not "depressed" [26]. It was further identified that stigma and related privacy concerns (i.e. concerns that members of their family or community would find out about their depressive

**Table 4. Summary of Findings– 4 key themes emerged regarding the conceptualization and experiences of PPD amongst refugee women resettled in high-income countries.**

| (1) Diverse cultural conceptualizations and experiences of postpartum depression | (2) Impact of mental health stigma on refugee women's experience of PPD & health-seeking behavior |
|---|---|
| ○ The term depression was reserved for extreme cases. Refugee women shared that Syrians experience less PPD due to positive cultural views of birth (26)<br>○ Bhutanese women lacked a cultural concept of PPD (30)<br>○ Karen refugee women viewed PPD as an alien experience for those who have a healthy baby (29)<br>○ Assyrian women had a clear understanding of PPD and were open to disclosure (29)<br>○ Emotional states influence how PPD is reported; somatic symptoms are common for displaying underlying distress (28, 31)<br>○ Non-Western cultures may lack framework for understanding tools that screen for PPD (28) | ○ Mental illness is perceived negatively in Syrian culture, and help-seeking may be impeded by privacy concerns and stigma from family members (26)<br>○ Stigma in Afghan culture towards mental illness perpetuated a perception that physicians would respond negatively to mental health issues (27)<br>○ Shame, stigma and an unwillingness to disclose emotional distress to HCPs prevents help-seeking in some refugee women populations (27, 29, 31)<br>○ Disclosure of negative feelings were often suppressed by fears of marital discord and fear of shame (28)<br>○ Bhutanese culture influences what terminology is stigmatized–"depression" was not stigmatized while "mental illness" was (30) |
| (3) Cultural traditions and social support play protective roles in postpartum mental wellbeing | (4) Impact of host culture on refugee women's experience of PPD |
| ○ Practicing cultural traditions and having intact social supports in one's host country are beneficial for postpartum emotional wellbeing<br>○ Previous social support from female kin and strong family bonds in one's native country are disrupted with forced migration contributing to significant stress for new mothers (26–28)<br>○ Traditional, religious and spiritual practices play a protective role in postpartum period (26, 27).<br>○ An absence of traditional practices (rest periods post-birth, and assistance from female kin) in one's host country is a source of tension and isolation for some refugee women (27, 30)<br>○ Supportive role of spouse in the peripartum period can differ in the country of resettlement from the spousal role in one's native culture (26, 27, 32) | ○ Western biomedical model of care tends to medicalize pregnancy and emphasize the individual when diagnosing and treating PPD (12)<br>○ Negative perception of refugee women in one's host country can contribute to feelings of alienation, displacement, and despair (12)<br>○ Variation amongst studies as to whether a shift from a collectivist to an individualistic culture was beneficial or harmful to the mental health of refugee women (26, 27, 30, 32)<br>○ Lack of experience of HCPs was described as a hindrance to achieving high-quality peripartum mental health care (31) though satisfaction with maternity care in host countries was described by some refugee women (27)<br>○ Refugee women from diverse cultures may not understand concepts included in items on the EPDS (28) |

symptoms if they disclosed them in a health care setting) were significant barriers to accessing mental health care amongst Syrian refugee women. Authors also found that due to this stigma, husbands may prevent their wives from disclosing depressive symptoms, further impeding help-seeking for mental health concerns in the postpartum period [26]. Given the cultural stigma around the term "depression", study participants suggested using the term "wellness" rather than "depression" when naming support programs or suggesting possible treatments [26].

Similarly, Afghan refugee women in the study by Russo et al. (2015) described the stigma in Afghan culture towards mental illness resulting in a reluctance to disclose mental health concerns to healthcare professionals [27]. This perception resulted in Afghan refugee women assuming that healthcare professionals would respond poorly to mental health issues ("think they were crazy") [27]. In this study, it was apparent that shame, stigma and an unwillingness to disclose emotional distress to healthcare professionals contribute to a delayed diagnosis of PPD, a significant concern also highlighted by Ussher et al. (2012), amongst Assyrian women, as well as by Asif et al. (2015) in their review of obstetric care of refugee and asylum-seeking women in the UK [27,29,31].

Stapleton et al.'s study (2013) which included refugee women from Somali, Sudan, Burundi, Afghanistan, and Liberia, further supported this view, finding that that traditional cultural beliefs linking mental illness to "sorcery, unavenged spirits, or curses" deter refugee women from disclosing pertinent information and seeking help [28]. For many women, disclosure of negative feelings is suppressed due to fears of causing marital discord and bringing shame to the family. Wanting to maintain strong social relations within family and community was therefore prioritized over disclosing or seeking help for mental health problems [28].

Davey (2013), in her examination of Bhutanese women's experience of PPD identified an interesting and unique paradox surrounding the stigmatization of mental illness in Bhutanese culture [30]. Although the term "mental illness" was stigmatized, the term "depression" was not [30]. This paradox was thought to arise from the distinction made in Bhutanese culture between brain-heart problems (where worries, anxiety, and depression are grounded) and brain-mind problems (in which mental illness is grounded) [30]. Brain-heart problems are seen as commonplace and normal, whereas brain-mind problems are stigmatized and rarely communicated [30].

**3. Protective role of cultural traditions and social support.** Another prominent theme that emerged is the important role cultural traditions and intact social supports play in the lives of refugee and asylum-seeking women particularly during the peripartum period. During forced migration and upon resettlement, there is a marked shift in access and exposure to the traditions and practices of one's native culture, as well as a significant disruption in social networks that are traditionally present around the time of childbirth. As described in several studies, practicing cultural traditions and having intact social supports in their host country are significantly protective for postpartum emotional wellbeing.

In many cultures, it is tradition to be surrounded by social support from female kin in the perinatal period [26,27]. This social support is completely disrupted by forced migration (28). In Ahmed et al's study, strong support provided around the time of birth was protective against depressive feelings and a lack of family support was a stressor in the postpartum period for several Syrian refugee women (2017) [26]. Afghan refugee women in Russo et al.'s study (2015) shared how a lack of connection to female kin and strong family bonds results in distress, loneliness and feelings of isolation in the postpartum period [27]. Refugee women in Stapleton et al.'s study (2013) reported the absence of close kin coupled with the loss of social supports contribute to PPD [28]. In support of this notion, Asif et al. (2015) highlighted that PPD tends to be more common in women who are both physically and culturally removed from their support systems [31].

Further, in the study by Ahmed et al. (2017), Syrian refugee women described the key role of engaging in other traditions and spiritual practices such as reading holy texts, like the Quran and praying in their new countries of resettlement [26]. Russo et al. (2015) highlighted that religion is intricately connected to culture for Afghan refugee mothers [27]. The importance of remaining connected to traditions, rituals and religion was described in several studies to protect women from experiencing PPD.

Davey (2013) reported how Bhutanese women shared the importance of maintaining traditions and rituals in the postpartum period along with having intact social support (30). In Bhutan, wellness is defined in the context of social ties and relationships and migration can dismantle these ties [30]. For instance, Bhutanese refugee women described a 1-month rest post-birth as a tradition in their culture [30]. During this time, women are considered "unclean" and are thus forbidden from partaking in their homemaking tasks. This gives the mother time to rest and recover in the postpartum period as well as an opportunity to connect with her baby [30].

Russo et al. (2015) described a similar cultural tradition for Afghan women of a 10 to 40-day postpartum rest period which includes assistance from a local birth attendant and female kin [27]. It concludes with ritual celebrations with traditional food. An absence of these traditional practices in their countries of resettlement was a source of tension and isolation for women in both studies [26,30]. Stapleton et al. (2013) reported similar findings in terms of rituals and ceremonies to welcome the newborn and celebrate the mother which are lacking when a pregnancy takes place in a new country of resettlement [28].

A few studies alluded to the supportive role of the spouse in the peripartum period that may differ in the country of resettlement from the spousal role in one's native culture [26,27,32]. In Russo et al. (2015), Afghan refugee women described the shifting role of their spouses during pregnancy and the peripartum period in their country of resettlement [27]. Without the typical female kin to support mothers in the postpartum period, husbands have to take on a greater role in providing emotional support for their wives compared to when they lived in Afghanistan [27]. In Ahmed et al. (2017), Syrian refugee women wondered whether their spouses would be prepared to take on the role of primary supporter in the perinatal period as they had not previously had such responsibility [26]. Nakash et al. (2016) provide another angle on this topic in their study examining the association between PPD, acculturation and the mother-infant bond among Eritrean asylum seekers in Israel [32]. As part of this study they asked women to rate the status of their current romantic relationship (using a 5-point Likert scale ranging from 1 "unhappy/unstable" to 5 "loving and stable") and also had women complete the Maternal Postnatal Attachment Scale (MPAS), a self-reported measure of mother-to-infant boding during the infant's first year of life [32]. What they found was that higher quality of current romantic relationship predicted a higher self-reported mother-to-infant bond [32]. This finding supports the notion that having a romantic partner that is perceived as loving and supportive is beneficial for refugee mothers and babies in their countries of resettlement and highlights the important role partners play in the peripartum period of refugee women at a time when social supports, including female kinship, are often lacking [32].

**4. Influence of host culture on the experience of pregnancy and postpartum depression.** Several articles discussed the significant impact the host culture has on the experience of pregnancy and postpartum depression in refugee women from diverse cultures, much of which stems from the Western biomedical model of care which tends to medicalize pregnancy and emphasize the individual when diagnosing and treating PPD.

The study by Brown-Bowers et al. (2015) explores this theme at length. The paper supports the view that the Western biomedical model of PPD, in which individual biological and psychological factors are emphasized, is particularly flawed when applied to refugee and asylum-seeking women from distinct cultures as it fails to account for the social, political and economic context of their symptoms and places responsibility on the individual to help or cure herself [12]. For example, depending on the political, economic and social climate of the host country during the time of resettlement, refugees may be "valorized or demonized" by the larger society. When refugees are positioned in particularly negative ways in their host countries, this creates an aversive environment for refugee women which may contribute to feelings of alienation, displacement, and despair [12].

Brown-Bowers et al. (2015) further discuss that the prevailing birth paradigm in Canada (which applies to most high-income countries) is dominated by technology and biomedicine where birth is a medical event controlled by highly trained medical specialists [12]. This paradigm may be jarring for women from different cultural backgrounds in which rich social networks, often led by experienced women from the community, guide and support women through pregnancy, birth, and the transition to motherhood. This can create a significant disconnect between the expectations of women's perinatal experience, informed by their native culture and past experiences, and their actual experience of care in their host country [12].

The shift to an individualistic culture was detrimental to Afghan refugee women in Russo et al.'s study (2015) [27]. Women in this study describe the feelings of isolation and loneliness in the postpartum period while highlighting the cultural hindrances to participating in activities that could alleviate social isolation. In Ahmed et al.'s study (2017), the Muslim refugee women were unable to partake in wellness activities that were open to both genders because they needed to cover their hair and body in the presence of men [26]. In this study, women

also described that non-pharmacological strategies (e.g. meditation, yoga) that are often proposed in Western culture to manage depression, are not well known or understood in Syrian culture serving as a barrier to participation and identifying an opportunity for more culturally-informed wellness interventions [26].

Davey (2013) provided a different perspective. One participant in this study shared that exposure to Canadian culture causes a shift from the collectivist concept of health present in Bhutanese culture to an individualistic view [30]. Davey describes that this shift may actually help women improve their ability to recognize individual feelings and express distress in the postpartum period in their new country of resettlement.

Nakash et al.'s study (2016) focused on the association between PPD, acculturation and the mother-infant bond among a group of Eritrean asylum-seekers using validated instruments including the Edinburgh Postnatal Depression Scale (EPDS), bicultural involvement, and adjustment scale and Maternal Postnatal Attachment Scale [32]. What they found was that higher levels of acculturation (to Israeli culture in this context) predicted both lower mother-infant attachment and higher EPDS scores. This highlights the potential for assimilation in one's host culture to negatively impact the emotional wellbeing of new mothers and their infants if it is done alongside limited involvement in one's own cultural practices.

A lack of experience of healthcare professionals was described as a hindrance to achieving care in the postpartum period by refugee women. Asif et al. (2015) recommended that healthcare providers must be educated in order to recognize that asylum-seeking women are high risk for the development of PPD [31]. Asif et al. (2015) identified both language and cultural differences as complicating the present lack of experience of healthcare providers. In contrast, Afghan refugee women in Russo et al.'s study (2015) were satisfied with the level of education and advice from maternal healthcare professionals [27]. They describe being happy with the maternity care received and highlighted the healthcare professionals as a source of emotional support. Brown-Bowers et al. (2015) add to this view stating that qualities like sympathy and friendliness, kindness, and support may be more important than having a care provider who has specific knowledge about a woman's cultural background [12].

Stapleton et al. (2013) provide a unique perspective on this topic, providing an in-depth exploration of the cross-cultural application of the EPDS, arguing that significant problems arise when it is applied as the standard screening tool for PPD in refugee women given its basis in a Western biomedical model of disease [28]. Refugee women from diverse cultures may not understand the concepts included in items on the EPDS, even with appropriate translation, as the transferability of many of the culturally specific concepts employed in the EPDS may be quite limited [28]. The variability in applications and interpretations of the EPDS in refugee women is highlighted by the fact that two studies included in this review used the EPDS with varying cut-off levels for a positive screen, and with distinct interpretations of the scale (e.g. in Ahmed et al. (2017), the EPDS was applied as a screening tool with a cut-off of 10 or more defined as a positive screen for being at risk for developing PPD [26]; in Nakash et al. (2016) the EPDS is applied with cut-off of 13 which is said to differentiate between minor and major depression [32], whereas Stapleton et al. (2013) outlines that an elevated score is 12 or more or a positive response to Item 10—self harm) [28].

## Discussion

Through this scoping review we sought to better understand what is currently known in the literature about the impact of culture on the conceptualization and experience of postpartum depression in refugee women as a distinct sub-population of migrant women. Four key themes emerged: 1) diversity in cultural conceptualizations and experiences of postpartum depression;

2) impact of mental health stigma on women's conceptualizations and experiences of postpartum depression and help-seeking behaviors; 3) protective role of cultural traditions and social support; and, 4) influence of host culture on the experience of pregnancy and postpartum depression.

It is interesting that despite this review including studies of women from markedly diverse countries and cultures (including Bhutan, Afghanistan, Syria, Burma, Somali, Liberia, and Burundi), clear commonalities in views and experiences of PPD emerged. What is more, the themes align with findings in the broader literature on PPD in migrant women more generally. This is made evident in the "meta-ethnographic" study conducted by Schmeid et al. (2017) on migrant women's experiences, meanings and ways of dealing with PPD which reviews 12 studies published between 1999 and 2016, all in high-income countries of resettlement [15]. In this review, it is striking how similar their findings are compared to our review which focused specifically on refugee women. In their paper, Schmeid et al (2017) highlight: 1) the notable stigma around mental illness which serves as a source of guilt, embarrassment and humiliation, and a barrier to disclosure and help-seeking for migrant women, 2) the trend towards somatization of emotions amongst migrant women; and, 3) the significant distress women experience at the loss of social supports and cultural traditions at this important time in their lives (15, 27). The similarity in findings across literature on PPD in migrant and refugee women stands in contrast to our hypothesis that there would be distinct differences between non-refugee migrants and refugees given the forced nature of their migration and associated stressors and trauma.

The thematic similarities found across the literature of PPD in migrant women post-resettlement raises the question of whether or not it is imperative that we stratify migrant women by migration type in studies on PPD, as has been suggested (and as we suspected) (17). Perhaps when it comes to looking specifically at the impact culture has on the conceptualization and experience of PPD, it is a woman's culture and its contrast with that of the culture of the country of resettlement that matters and not her migration status. This is plausible. However, we would caution against assuming that stratification by migrant type is not of value for three key reasons. Firstly, our review identified only 8 studies representing approximately 300 refugee and asylum-seeking women resettled in high-income countries around the world. To make any final conclusion on PPD in refugee populations with such a limited evidence base would be premature. Secondly, a number of studies of PPD amongst migrant women included refugee women but made no distinction between these migrant populations making it difficult to know which migrant sub-type was driving the findings. Stratification would have shed light on this. Thirdly, there are likely other key factors beyond culture that impact refugee women's conceptualization and experience of PPD, which have not been explored through this scoping review. Such factors include duration of migration journey, time spent in a neighboring low- or middle-income country or refugee camp before resettling in their host country, and time since arrival in one's host country. The sociopolitical and economic climates of a woman's country of origin and host country may also play an important role. Research that distinguishes between migrant type (i.e. economic or family migrant vs. refugee or asylum-seeker) and better characterizes the migration journey of women is needed before any conclusion can be made about how to best stratify data on migrant women in research on the topic of PPD.

One unexpected finding in our review which has received little attention elsewhere is the significant shift in role and responsibility in the peripartum and postpartum periods experienced by the male partners of the women included in the reviewed studies. This was characterized as a shift from having less engagement in their home country during the peripartum and postpartum periods due to the central supportive role of female kinship to being the central support for their wives in their country of resettlement given the absence of the usual support systems of female kinship. This is an interesting finding as it highlights the importance of

ensuring male partners are not forgotten in the peripartum and postpartum care of refugee women in high-income countries. Ensuring that culturally sensitive strategies are in place that include men in the care of their wives and newborn children is imperative to ensure they are supported in their transition to this new and perhaps unfamiliar role.

In terms of health care service provision, health care providers that care for refugee women may feel overwhelmed by the diversity of cultural conceptualizations and experiences outlined in this review. It should be emphasized that it is not the responsibility of health care providers to know about each different culture specifically, as that would be an untenable proposition. Instead it is suggested that HCPs approach the topic of emotional wellbeing of refugee women in the postpartum period with an awareness that their conceptualization of emotional wellness and distress is informed by their cultural background and may be distinct from the Western biomedical model of health and mental illness. Being curious and asking open-ended questions serves as a good starting point. Being aware that stigma towards mental illness may be a significant barrier to disclosing emotional distress postpartum, as highlighted in this review, can guide one's choice of language to using terms such as "wellness" and "supports" instead of "depression" or "mental health concerns". Taking time to inquire about social supports, cultural traditions around pregnancy, and how engaging in them could be facilitated in the country of resettlement may go a long way in positively impacting the emotional wellbeing for those women who may be struggling. This point is emphasized by Tobin et al. in their 2017 meta-synthesis of refugee and immigrant women's experience of PPD, in which authors state that: "Providers who understand the cultural implications of PPD have greater insight into the unique needs of the mothers in their care, which builds trust between the providers and their clients. This in turn is more likely to result in effective care and treatment" [14].

Further, using the EPDS to screen for PPD (and moving on if it is negative), is insufficient and arguably inappropriate when applied to the care of refugee women. Health care providers should reflect on their current use of this tool to ensure it is being used with an awareness of its shortcomings (e.g. cultural and linguistic misunderstandings in regard to certain EPDS items). Stapleton et al.'s paper (2013) reviewed here can serve as an informative guide to this end [28].

This review also highlights that it is important to remember that just as refugee women are embedded in their own cultural backgrounds which inform their conceptualizations and experiences of PPD, so are all health care providers. For practitioners who are native to the high-income countries in which they practice (or have acculturated since their own arrival), this is often a cultural view that endorses individualism and a biomedical model of disease. Here, individualism can be defined as "a social pattern that consists of loosely linked individuals who view themselves as independent of collectives; are primarily motivated by their own preferences, needs, rights . . . [they] give priority to their personal goals over the goals of others; and emphasize rational analyses of the advantages and disadvantages to associating with others." [33]. This cultural framework may stand in stark contrast to the views of the collectivist cultures of many of refugee women which are defined by "a social pattern consisting of closely linked individuals who see themselves as parts of one or more collectives . . . [they] are primarily motivated by the norms of, and duties imposed by, those collectives . . ." [33]. Therefore, it is not just for us as health care providers to acknowledge the cultural perspective of refugee women but also to reflect and critically explore our own cultural perspectives and how these views may contribute to the challenges faced by refugee women upon resettlement. Engaging in cultural competency training during medical school or through continuing professional development for health care providers already in practice in one way that this critical refection and exploration could be achieved.

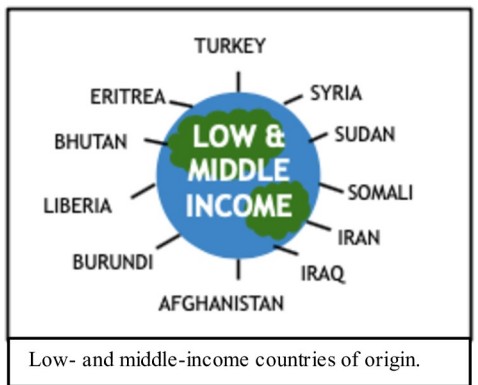
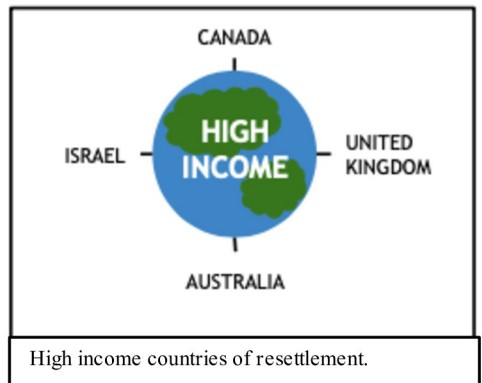

| Low- and middle-income countries of origin. | High income countries of resettlement. |

**Fig 2. Refugee women's countries of origin (low- and middle-income) and countries of resettlement (high-income) included in scoping review.**

Our study findings must be interpreted within the limitations of the methodology applied. Due to our criteria that articles include an explicit sub-analysis of refugee women's cultural conceptualization and/or experience of PPD, studies that included refugee women under the broader categorization of "migrant" or "immigrant" women were excluded. We chose this approach as such studies did not provide findings stratified by migrant type, making it impossible to extract data from these studies that was specific to refugee women. Further, our search strategy specified that articles use the term "refugee" (and analogous terms as outlined in our search strategy in Table 2). This may have inadvertently excluded studies of women who fit the definition of "refugee" if this specific term or analogous terms were not explicitly used. This question arose, for example, when screening studies of Hispanic and Latino women resettled in the United States, in which it was not always clear (and often not explicitly stated) whether they were refugee women or not [34,35]. It should also be recognized that the included studies represent a very small sample size (a total of approximately 300 women) from a limited number of countries and resettling in only 4 high-income countries (Fig 2). Notably, studies from the United States and Germany (countries that resettle numerous refugees) that fit our inclusion criteria were lacking [4,5]. Further research is needed to better understand the broad array of conceptualizations and experiences of PPD held by refugee women from and resettling in diverse countries and cultures around the world which are not yet represented in the literature. This includes a closer examination of the impact of culture on the conceptualizations and experiences of PPD of refugee women resettled in low- and middle-income countries which were excluded from this review. Lastly, as mentioned, there are other important factors beyond culture which likely impact refugee women's conceptualization and experience of PPD that have not been explored through this scoping review such as duration of migration journey, time since arrival in host country and the sociopolitical and economic climates of a woman's country of origin and host country.

## Conclusion

The studies identified through this scoping review provide a rich description of the significant impact culture has on how refugee women conceptualize and experience postpartum depression when resettled in high-income countries. The highlighted themes align with those seen in the literature on migrant women in general, however, there remains insufficient evidence regarding whether significant differences exist in how culture impacts refugee women's experiences of PPD compared to migrant women of non-refugee backgrounds. More in-depth

studies that assess duration of the migration journey, time in refugee camps, time since arrival in host country and sociopolitical and economic climate are needed. With this, we can better understand how the experiences of migrant women with refugee and non-refugee backgrounds in this area of research.

Health care providers caring for refugee women in the postpartum periods should acknowledge and explore the cultural implications of PPD in order to appropriately assess, diagnose and treat PPD in refugee women from diverse cultural backgrounds. Only then will we be able to achieve optimal mental health outcomes for newly postpartum refugee mothers who are increasingly resettling in high-income countries around the world.

## Supporting information

**S1 Checklist. Preferred Reporting Items for Systematic reviews and Meta-Analyses extension for Scoping Reviews (PRISMA-ScR) checklist.**
(PDF)

**S1 Table.**
(DOCX)

**S2 Table.**
(DOCX)

## Author Contributions

**Conceptualization:** Saarah Haque, Mary Malebranche.

**Formal analysis:** Saarah Haque, Mary Malebranche.

**Methodology:** Saarah Haque, Mary Malebranche.

**Supervision:** Mary Malebranche.

**Writing – original draft:** Saarah Haque.

**Writing – review & editing:** Saarah Haque, Mary Malebranche.

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
