## [Decision Letter · Decision Letter 0]

18 Jun 2020

PONE-D-19-29557

Impact of culture on refugee women's conceptualization and experience of postpartum depression in high-income countries of resettlement: A scoping review

PLOS ONE

Dear Dr. Malebranche,

Thank you for submitting your manuscript to PLOS ONE. After careful consideration, we feel that it has merit but does not fully meet PLOS ONE’s publication criteria as it currently stands. Therefore, we invite you to submit a revised version of the manuscript that addresses the points raised during the review process.

We look forward to receiving your revised manuscript.

Kind regards,

Kyoung-Sae Na, M.D.

Academic Editor

PLOS ONE

Journal Requirements:

2. We ask that you please complete and upload the PRISMA-ScR checklist as Supporting Information for this study. The checklist can be found here: http://www.equator-network.org/wp-content/uploads/2018/09/PRISMA-ScR-Fillable-Checklist.pdf.

Thank you for your attention to this request.

Reviewers' comments:

Reviewer's Responses to Questions

**Comments to the Author**

1. Is the manuscript technically sound, and do the data support the conclusions?

Reviewer #1: Yes

Reviewer #2: Yes

Reviewer #3: Partly

2. Has the statistical analysis been performed appropriately and rigorously? 

Reviewer #1: N/A

Reviewer #2: N/A

Reviewer #3: Yes

3. Have the authors made all data underlying the findings in their manuscript fully available?

Reviewer #1: Yes

Reviewer #2: Yes

Reviewer #3: Yes

4. Is the manuscript presented in an intelligible fashion and written in standard English?

Reviewer #1: Yes

Reviewer #2: Yes

Reviewer #3: Yes

5. Review Comments to the Author

Reviewer #1: The authors undertook a scoping review to assess the impact of culture on the conceptualization and experience of postpartum depression (PPD) by refugee or asylum seeking women resettled in developed countries. The PRISMA guidelines for scoping reviews has been followed.

The search terms "postpartum depression" in Table2 is repeated

why was the review limited to only developed countries? Can the authors provide more justification for the choice?

Reviewer #2: Hypothesis: There are important differences in the conceptualization and experiences of PPD between refugee women and other migrant women which have yet to be clearly defined.

Objective: to characterize what is currently known in the literature about the impact of culture on the conceptualization and experience of PPD in refugee women resettled in high-income countries.

The authors undertook a scoping review justifying that it is the preferred methodological framework due to the expected heterogeneity in the current research disciplines and methodologies addressing this question.

Overall, the manuscript is well written. However, it could benefit with concision, including reporting some of the key findings diagrammatically or in table, if possible.

Some specific comments are mentioned below.

Abstract

Line 48: may be omit “will”

Briefly report the eligibility criteria for including studies

Introduction

Introduction is written well and provides strong rationale for the scoping review.

Line 115-116: It is not clear what the following statement means. “However, a focus on the sub population of refugee women is often lacking.”

Line 127: Need reference.

Line 128-131: Please consider revising. I would say, we assessed impact of culture on the conceptualization and experience of PPD in refugee women, and what are the implications for diagnosing and managing PPD…

Methods

Line 140-143: I think these lines are redundant. Objectives are rationale are covered in the Introduction.

Line 167-168: Please consider revising. “was their” is not going well.

Results

293-295: should this be part of theme 3?

318-383: provide reference(s)

385-386: provide reference(s)

Limitations

It seems, refugees in LMICs were excluded, assuming that the culture may not be different from the host country. This may be not entirely true. Consider mentioning this in Limitations.

Reviewer #3: This scoping review is a needed contribution within refugee maternal mental health scholarship. Please take a look at what JBI says regarding the intentions of a scoping review to align with your goals/objectives. I've made some other recommendations and wish you all the best in your work with refugee women. Thank you for bringing to light the need for more inquiry within this often overlooked population.

6. PLOS authors have the option to publish the peer review history of their article (what does this mean?). If published, this will include your full peer review and any attached files.

Reviewer #1: No

Reviewer #2: No

Reviewer #3: Yes: Shahin Kassam, RN, PhD(c)

---

## [Author Response · Author response to Decision Letter 0]

3 Jul 2020

Dear PLOS ONE Academic Editor and Peer Reviewers,

Many thanks for your thorough review of our manuscript entitled "Impact of culture on refugee women's conceptualization and experience of postpartum depression in high-income countries of resettlement: A scoping review" (PONE-D-19-29557). We greatly appreciate all the feedback, comments and questions you provided. We have taken the time to work through them and have provided responses to each in the attached "Responses to Reviewers" document. We have also submitted a revised version of our manuscript with track changes and another without as per your request. A PRISMA checklist for scoping reviews has also been completed and uploaded. 

We look forward to hearing from you once you have had a chance to review our resubmission. Your consideration of our manuscript for publication in PLOS ONE is greatly appreciated. 

Kind regards,

Dr. Mary Malebranche

---

## [Decision Letter · Decision Letter 1]

11 Aug 2020

Impact of culture on refugee women's conceptualization and experience of postpartum depression in high-income countries of resettlement: A scoping review

PONE-D-19-29557R1

Dear Dr. Malebranche,

We’re pleased to inform you that your manuscript has been judged scientifically suitable for publication and will be formally accepted for publication once it meets all outstanding technical requirements.

Kind regards,

Kyoung-Sae Na, M.D.

Academic Editor

PLOS ONE

Additional Editor Comments (optional):

Reviewers' comments:

Reviewer's Responses to Questions

**Comments to the Author**

1. If the authors have adequately addressed your comments raised in a previous round of review and you feel that this manuscript is now acceptable for publication, you may indicate that here to bypass the “Comments to the Author” section, enter your conflict of interest statement in the “Confidential to Editor” section, and submit your "Accept" recommendation.

Reviewer #2: All comments have been addressed

Reviewer #3: All comments have been addressed

2. Is the manuscript technically sound, and do the data support the conclusions?

Reviewer #2: Yes

Reviewer #3: Yes

3. Has the statistical analysis been performed appropriately and rigorously? 

Reviewer #2: N/A

Reviewer #3: N/A

4. Have the authors made all data underlying the findings in their manuscript fully available?

Reviewer #2: Yes

Reviewer #3: Yes

5. Is the manuscript presented in an intelligible fashion and written in standard English?

Reviewer #2: Yes

Reviewer #3: Yes

6. Review Comments to the Author

Reviewer #2: The authors have made several changes to the manuscript and have improved it significantly. Best wishes.

Reviewer #3: (No Response)

7. PLOS authors have the option to publish the peer review history of their article (what does this mean?). If published, this will include your full peer review and any attached files.

Reviewer #2: No

Reviewer #3: **Yes: **Shahin Kassam

---

## [Editor Report · Acceptance letter]

18 Aug 2020

PONE-D-19-29557R1 

Impact of culture on refugee women's conceptualization and experience of postpartum depression in high-income countries of resettlement: A scoping review 

Dear Dr. Malebranche:

I'm pleased to inform you that your manuscript has been deemed suitable for publication in PLOS ONE. Congratulations! Your manuscript is now with our production department. 

Kind regards, 

on behalf of

Dr. Kyoung-Sae Na 

Academic Editor

PLOS ONE